# Optimum Post-Discharge Care of Acute Kidney Injury (AKI) Survivors

**DOI:** 10.3390/jcm11216277

**Published:** 2022-10-25

**Authors:** Abdulrahman Alwagdani, Alaa S. Awad, Emaad M. Abdel-Rahman

**Affiliations:** 1Division of Nephrology, University of Virginia, Charlottesville, VA 22903, USA; 2Division of Nephrology, University of Florida, Jacksonville, FL 32209, USA

**Keywords:** AKI, AKI-D, management, renin−angiotensin system inhibitors

## Abstract

Acute kidney injury (AKI) is a serious problem, affecting multiple organs, and is associated with a high mortality. The severe consequences of AKI extend beyond hospital discharge to the outpatient setting. While a plethora of literature exists guiding the management of AKI in the hospital setting, currently, there are no guidelines for the best care of AKI patients post-hospital discharge. In this review, we address the burden of AKI on patients and the importance of optimal coordinated care of these patients post-hospital discharge. We review the care of patients with or without dialysis requirements at the time of discharge and thereafter.

## 1. Introduction

The Kidney Disease: Improving Global Outcomes (KDIGO) guidelines define acute kidney injury (AKI) as an abrupt decrease in kidney function that occurs over a period of 7 days or less [1] in contrast to chronic kidney disease (CKD), defined as abnormalities in kidney structure or function that persist for >90 days [2]. AKI results in a reduction in the glomerular filtration rate (GFR) and an accumulation of waste products in the blood with or without decreased urine output. The diagnostic criteria of AKI have been developed for the identification and monitoring of AKI [1].

AKI is common, affecting up to 1% of the general population and 8–20% of hospitalized patients; it was shown to complicate 30–50% of critical care admissions [3,4,5,6,7,8,9]. The rate of AKI has been steadily increasing despite improved diagnostic criteria and early AKI recognition [8]. Several studies reported an increased incidence of patients hospitalized with AKI [7,9]. In a systemic review that included 49 million patients, mostly from high-income countries, 1 in 5 adults hospitalized with acute illness had AKI [4].

AKI is associated with serious consequences, both while in the hospital and after discharge. While an early recovery of AKI can occur in patients in the hospital setting, mortality and morbidity remain very high, especially in its most severe form: AKI requiring dialysis (AKI-D) [10,11]. Other complications of AKI noted in the hospital setting include fluid overload, electrolyte abnormalities, acid base imbalance, and infections. 

The post-hospital discharge of patients with AKI can have short- and/or long-term sequela. Patients with AKI are at a higher risk of progressively losing their kidney function up to end-stage kidney disease (ESKD). In a meta-analysis by Coca et al. of 13 cohort studies, patients with AKI had higher risks for developing chronic kidney disease (CKD), with a hazard ratio (HR) of 8.8 and a 95% confidence interval (CI) of 3.1–23.5, compared with patients without AKI [12]. A transition period between AKI and CKD was termed acute kidney disease (AKD). In a consensus statement, Chawla et al. proposed staging criteria for AKD and suggested strategies for the management of these patients [10]. AKI affects other organ systems; cardiac (congestive heart failure, coronary artery disease, and stroke), pulmonary, hepatic, and bone diseases may follow. Grams and Rabb reviewed clinical and preclinical data that highlighted the pathways through which AKI affected these distant organs [13].

On the other hand, AKI patients discharged from the hospital with AKI-D may recover, progress to ESKD, or die. In our center, we followed a cohort of AKI-D patients short-term (up to 12 weeks) and long-term, from after week 12 (median 859.7 days with a range of 7–1394 days) [14,15]. We showed that recovery, defined as being dialysis-independent, occurred in 42.0% and 35.2% over the short- and long-term periods, respectively. Death occurred in 9.2% and 29.6% over the short- and long-term periods, respectively. Patients with end-stage kidney disease (ESKD) was declared in 48.7% and 35.8% over the short- and long-term periods, respectively. ESKD was defined as those requiring hemodialysis (HD) for more than 90 days. It is important to note that only 3/52 (5.8%) patients recovered enough kidney function to become dialysis-independent after the first 12 weeks. This data highlights the importance of taking adequate care of patients with AKI surviving hospital admission. 

Currently, there is no universal definition of a post-hospitalization AKI survivor; however, we postulate that a post-hospitalization AKI survivor is a person discharged alive from a hospital following an episode of AKI whether on or off dialysis. 

## 2. Complications of AKI

### 2.1. AKI Mortality

The mortality of patients with AKI can occur early while in the hospital or later post-hospital discharge. In-hospital mortality remains extremely high among patients with AKI and is more prominent in patients with AKI-D [7,8,16]. In a recent retrospective cohort study using data from the national Veterans Health Administration examining in-hospital and 1-year mortality among patients with AKI, 6% of patients died in the hospital with 28% dying within 1 year, compared to 0.8% in-hospital and 14% 1-year mortality in patients without AKI [17].

### 2.2. AKI Morbidity

#### 2.2.1. AKI and the Cardiovascular System

AKI is associated with an increased risk of cardiovascular mortality and morbidity. A systemic review done by Odutayo et al. that included 25 studies involving 254,408 adults, of whom, 22% had AKI, found that AKI is associated with an 86% and a 38% increased risk of cardiovascular mortality and major cardiovascular events, respectively (RR 1.86; 95% confidence interval (95% CI), 1.72 to 2.01) and (RR 1.38; 95% CI, 1.23 to 1.55), respectively) [18]. For disease-specific events, AKI was shown to be associated with a 58% increased risk of heart failure with a relative risk (RR) of 1.58; 95% CI, 1.46 to 1.72 and a 40% increased risk of acute myocardial infarction (MI) (RR 1.40; 95% CI, 1.23 to 1.59). AKI was also associated with a 15% increased risk of stroke (RR 1.15; 95% CI, 1.03 to 1.28) [18]. Furthermore, within five days after elective cardiac surgery, postoperative AKI was associated with an increased five-year mortality and a statistically insignificant increased risk of MI [19].

Associations between AKI and long-term outcomes, including mortality, ESKD, and cardiovascular and renal hospitalization, were studied in patients who received coronary angiography and showed an increased adjusted risk of death, progression to ESKD, and subsequent hospitalization for cardiovascular and renal events with increasing severity of AKI, although the gradient of risk across stages of AKI differed among these events [20].

Hypertensive patients with postoperative AKI have a significantly higher risk of fatal stroke and fatal MI as well as all-cause mortality within 5 years after elective non-cardiac surgery [21]. Similarly, patients receiving angiotensin-converting enzyme inhibitors (ACEI)/angiotensin receptor blocker (ARB) and calcium channel blockers (CCB) in the perioperative period showed a significant association with fatal stroke and MI post-operatively [21]. Furthermore, patients who recovered from AKI had a higher incidence of developing incidental stroke and increased mortality than the patients without AKI; this impact was similar to the impact of diabetes [22].

#### 2.2.2. AKI Recurrence and Progression to CKD & ESKD

AKI progressing to CKD and AKI progressing to ESKD is common in AKI survivor patients [10]. In a study done on 11,683 patients by Siew et al. on the clinical risk factors for recurrent AKI within 12 months, they showed that 25% of these patients had recurrent AKIs [23]. Older age was associated with the highest risk of recurrent AKI. Discharge diagnoses most strongly associated with recurrent AKI included the following: congestive heart failure (CHF), decompensated liver disease, malignancy (with or without chemotherapy), acute coronary syndrome, and volume depletion. Lower inpatient serum albumin level and CKD showed a dose-dependent association with recurrent AKI. Interestingly, the severity of the AKI was not associated with the risk for recurrent AKI [23]. These factors should be considered when deciding on what is the optimum time for follow-up after hospital discharge.

The use of angiotensin-converting enzyme inhibitors/angiotensin receptor blockers (ACEI/ARBS) in patients with AKI during the index hospitalization was not associated with a higher risk of recurrent hospitalized AKI (adjusted hazard ratio, 0.88; 95% confidence interval, 0.69 to 1.13) [24].

An episode of AKI is associated with ultrastructural changes in the renal pathology. Animal kidney models with AKI showed fibrosis, vascular infarction, tubular loss, glomerulosclerosis, and chronic interstitial inflammation, which ultimately leads to kidney aging, slow decline in kidney function, and, consequently, to CKD [25]. Wald and colleagues showed that patients with AKI who required inpatient dialysis had an increased rate of developing CKD [26]. Even patients with resolved AKI showed that the disease may still be complicated by de novo CKD. In a study on 1610 patients with resolved AKI within 90 days of hospital discharge, who had a follow-up period of 3.3 years, a resolved AKI was associated with a significant risk of de novo CKD (hazard ratio 1.91) [27]. A matched cohort of 1538 hospitalized adults with and without AKI in the ASSESS-AKI study were followed for up to 3 months after discharge to assess the effect of serum and urine albumin on kidney function prognosis. This study showed that a higher urinary albumin to creatinine ratio (UACR) post-AKI was associated with an increased risk of kidney disease progression (hazard ratio 1.53 for each doubling; 95% CI, 1.45–1.62), and the urine albumin measurement was a strong discriminator for future kidney disease progression (C statistic, 0.82) [28]. 

#### 2.2.3. AKI and Cost

In the United States, AKI is associated with an increase in hospitalization costs that range from USD 5.4 to 24.0 billion annually. The average cost of hospital stays involving AKI was nearly double the cost of stays without AKI. In 2014, the average cost for a hospital stay that involved a diagnosis of AKI (USD 19,200/patient/hospitalization) was nearly twice as high as for stays that did not involve AKI (USD 9900/patient/hospitalization) [29,30]. These complications highlight the important care for AKI survivors; it should be optimized to prevent/minimize these complications.

## 3. Care for AKI Survivors: (Table 1)

Caring for AKI survivors can be divided into inpatient and outpatient or post-discharge care.

**Table 1 jcm-11-06277-t001:** Intervention for AKI survivors.

Determinants of AKI-D Outcomes	Intervention
- **Education**	Education of patient/caregiver and team members about better care of the kidney post-hospital discharge.
- **Co-morbidities:** Severity of acute disease Pre-existing CKD Prior AKI Diabetes mellitus (DM) Cardiovascular diseases (CVD) Volume overload Other co-morbidities: chronic liver diseases, cancer, surgeries.	Adequate management of DM, CVD, volume overload, and other comorbidities.
- **Laboratory parameters**	Monitoring serum creatinine, cystatin C, and proteinuria for renal recovery.
- **Medications:** Renin−angiotensin system inhibitors (RASi) Diuretics Others	- Restarting RASi when kidney function returns to baseline/new baseline. - Diuretics help with volume management and may help in kidney recovery following AKI. - Adjusting dose of medications based on kidney function.
- **Outpatient care of patients** **Dialysis-related factors** Renal replacement modality Dose and duration **of renal replacement therapy** Intradialytic hypotensive episodes **Nephrology post-AKI follow-up**	For AKI-D: - Avoid frequent hypotensive episodes and limit ultrafiltration. For both AKI and AKI-D - Communication among different team members (nephrologist, primary care provider, pharmacists, nurses, and other relevant subspecialties).
- **Biomarkers** urinary C−C motif chemokine ligand 14 (uCCL-14) urinary neutrophil gelatinase-associated lipocalin (uNGAL)/creatinine ratio urinary ganglioside M2 activator protein (uGM2AP) tail-less complex polypeptide-1 eta subunit (uTCP1-eta)	- Limited data available identifying biomarkers predicting further AKI episodes.

CKD: Chronic Kidney Disease; AKI: Acute Kidney Injury; DM: Diabetes Mellitus; CVD: Cardiovascular Disease; RASi: Renin Angiotensin System Inhibitors; AKI-D: Dialysis Requiring Acute Kidney Injury; uCCL-14: Urinary C−C motif chemokine ligand 14; uNGAL: Urinary neutrophil gelatinase-associated lipocalin; uGM2AP: Urinary ganglioside M2 activator protein; uTCP1-eta: tail-less complex polypeptide-1 eta subunit.

### 3.1. Inpatient AKI Survivors’ Care

Caring for inpatient AKI requires a multidisciplinary approach to optimize and maintain AKI care while hospitalized. Reviewing medications, hemodynamics, volume status, urine output, and other parameters that can affect kidney function should be assessed daily. An early nephrology consult can help in optimizing care for AKI patients. A meta-analysis done by Soares et al. showed that a delayed nephrology consultation was associated with higher mortality in AKI patients [31]. Details of inpatient care of patients with AKI is beyond the scope of this review.

### 3.2. Care of AKI Survivors Post-Hospital Discharge 

#### 3.2.1. General

When the AKI patient is ready for hospital discharge, formal planning should be formulated by a multidisciplinary team. Studies found that AKI survivors have poor knowledge and awareness of AKI and, subsequently, a higher risk for rehospitalization within the first 30 days of discharge [3,32]. Educating patients and caregivers about optimum kidney care by the primary team/nephrologist or the nursing staff should be provided. Education should be continued in the subsequent clinic visits.

Reviewing discharge medications by the pharmacists and providing an adequate discharge summary by the primary team are crucial for further post-discharge care. A discharge summary should include the cause of AKI; hospital course; details about intensive care admission and dialysis if utilized in the inpatient setting; medications, including the need of vasopressors; and a detailed plan for follow-ups as to when, where (specialized clinic, kidney clinic, rehabilitation facility, or a dialysis unit for AKI-D patients), and how frequently the patient should be followed. A follow-up with a multidisciplinary team that includes a nephrologist, primary care provider (PCP), pharmacist, and nutritionist should be recommended with adequate communication among members of the team for all AKI patients, especially those patients who are at high risk for recurrent AKI, worsening kidney function, or cardiac complications. Monitoring for predictors of outcomes of AKI needs to be routinely done in the outpatient setting [33].

#### 3.2.2. AKI-D Patients

The discharging team will be tasked with arranging for dialysis (whether permanent or temporary) before the patient’s discharge with adequate communication between the discharge team and the nephrologists accepting the patient at their unit. It is important to clearly indicate that the patient is diagnosed with AKI, rather than ESKD. 

##### Patient-Centered Care

A prolonged discussion between providers, patients, and caregivers about the patients’ dialysis options, the need to care about their vascular access, and being compliant with dietary and fluid recommendations as well as dialysis appointments needs to take place. Supplying patients and caregivers with educational materials can be very helpful. A dialysis prescription needs to be individualized with frequent follow-ups by the nephrology team. It is important to check pre-dialysis weekly laboratory data. Weekly laboratory analysis includes obtaining the basic metabolic panel (BMP) that includes the renal function tests and electrolytes and 24 h urine (in non-oliguric patients) to check urea and creatinine clearances. If the patient’s average urea and creatinine clearance is above 15 mL/minute, we hold HD and check BMP at the time of the hemodialysis session. If the patient is stable clinically with no rise in serum creatinine for a week, we arrange an outpatient clinic follow-up. If the patient remains clinically stable with stable/improved renal function, the dialysis catheter is removed, and the patient continues to follow up with the outpatient kidney clinic.

##### Potential Interventions

Avoiding excessive ultrafiltration (UF) and intradialytic hypotension (IDH) may favor the outcome. In a recent study, Abdel-Rahman et al. studied all patients with AKI-D discharged to outpatient HD units between January 2017 and December 2019 (n = 273) and followed them for up to a 6-month period [34]. Dialysis-related parameters were measured during the first 4 weeks of outpatient HD. They showed that more frequent IDH episodes were associated with an increased risk of ESKD (*p =* 0.01). They further demonstrated that a one-liter increment in net UF was associated with a 54% increased ratio of ESKD (*p* = 0.048). Pajewski et al. showed similar results [35]. These data highlight the importance of optimizing dialysis prescription to decrease the frequency of IDH episodes and minimize UF along with close monitoring of outpatient dialysis in patients with AKI-D. This ultimately will improve outcomes for such patients.

#### 3.2.3. AKI Patients Not Requiring Outpatient Dialysis

Outpatient or post-discharge AKI care can be broadly divided into four targets: patient-centered outcomes, potential interventions, and intermediate- and long-term clinical outcomes. 

##### Patient-Centered Outcomes

Patient-centered outcomes include an assessment of the symptom burden, such as fatigue, weakness, and anorexia; an assessment of the psychological burden, such as depression, health-related quality of life, and frailty; and education and awareness about kidney disease. KDIGO guidelines on AKI recommends a follow-up after 3 months post-discharge. However, this should be individualized based on the severity of AKI, comorbid conditions, and age. Utilizing tele-health visits can be considered in certain patients. Deciding on which patient should follow up with nephrology in the outpatient settings is not clear. We suggest that patients with a high risk of recurrent AKIs (older age, CHF, advanced liver disease, malignancy, acute coronary syndrome, malnutrition, or >2 comorbid conditions), eGFR < 60, severe AKI that has not returned to baseline before discharge, and AKI-D survivors be seen by a nephrology service in the outpatient settings. For other patients, they can be seen by their PCPs and referred to nephrology if they have residual CKD after the resolution of AKI and/or developed recurrent AKI. A recent study showed a lower all-cause mortality of 8.4% vs. 10.6% for AKI-D survivors who were seen by a nephrologist within 3 months of hospital discharge compared to AKI-D survivors who did not follow up with nephrology [36].

##### Potential Interventions

Each visit should assess risks of the progression of kidney disease, initiate or reinitiate kidney protective medications (e.g., ACEI/ARB), monitor for nephrotoxic medications, detect and reduce risk for recurrent AKI, and improve clinical follow-up and rehabilitation if needed. 

Brar et al. examined the role of ACEI/ARBs in improving the outcomes of AKI patients post-hospital discharge. They retrospectively analyzed the data of 46253 adults with AKI post-hospital discharge, they followed them up to 2 years, and showed that receiving these medications was associated with a lower mortality but a higher risk of kidney-related hospitalization [37].

Monitoring kidney function is traditionally done by measuring serum creatinine and blood urea nitrogen; however, this can be misleading for patients with low muscle mass. Monitoring cystatin C or 24 h urine for urea and creatinine clearance may be a better option for such patients. 

New biomarkers, such as liver-type fatty acid-binding protein (u-LFABP), neutrophil gelatinase-associated lipocalin (NGAL), connective tissue growth factor (CTGF), and Interleukin 18 (IL-18), provide a promising future as AKI biomarkers for kidney function. However, they are still under investigation and not nationally available [38,39]. In addition, their role is still limited in the clinical settings. Monitoring signs of uremia in patients with advanced CKD or AKI survivors that have not recovered fully from the AKI as well as early referral for transplant and discussion about different modalities for renal replacement therapy vs. palliative care should be considered once the patient’s kidney function has stabilized for few months after the AKI. Restarting or prescribing medications after AKI is an important part of the post-hospital follow-up. Renin−angiotensin−aldosterone system inhibitors (RAASi) are usually stopped during AKI together with blood pressure medications and diuretics. Restarting these medications should be considered as needed [40,41,42,43,44]. In a study of AKI survivors attending AKI clinics, RAASi are commonly discontinued in the setting of hospitalized AKI, and acute exposure to RAASi during hospitalization does not appear to increase the risk of persistent kidney dysfunction at 3 months post-discharge. Similar results were noted among survivors with and without AKI during hospitalization in the ASSESS-AKI study where patients exposed to RAASi were evaluated at 3 months post-discharge. Exposure to RAASi was not associated with a higher risk of recurrent hospitalization with AKI, death, kidney disease progression, or heart failure events during a median follow-up of 4.9 years [45].

##### Intermediate and Long Care Outcomes

Intermediate care outcomes include monitoring for proteinuria; chronic diseases, such as hypertension and diabetes mellitus; rehospitalization; recurrent AKIs; and adverse drug events, while long-term outcomes include the monitoring of CKD and cardiovascular risk factors. Assessing kidney function recovery and proteinuria status 3 months after AKI provides important prognostic information for long-term clinical outcomes, as shown in the ASSESS-AKI study [28]. Complications of kidney disease, including hyperphosphatemia, hyperparathyroidism, low vitamin D, and anemia, should be treated as needed to help improve the quality of life of these patients as well as help avoid further complications. Various prediction tools have been developed for predicting AKI in different clinical settings (e.g., ICU and cardiovascular surgery). Although not 100% accurate, these prediction tools can be used to identify patients who have a higher risk of AKI and establish preventive measures to decrease the AKI risk [40,46,47].

## 4. Conclusions

AKI is associated with high mortality, morbidity, and health care expenditure. Optimum care for AKI survivors requires a multidisciplinary approach, which should be started while patients are hospitalized and continue as outpatients. Care should be individualized depending on patient’s demographics, risk of recurrence, and severity of AKI and AKI-D. Spectrums of care include patient-centered outcomes, potential interventions, and intermediate- and long-term clinical outcomes. 

## Data Availability

Not applicable.

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
