# Peer review of "Optimum Post-Discharge Care of Acute Kidney Injury (AKI) Survivors"

_jcm, 2022, doi:10.3390/jcm11216277_

Round 1
Reviewer 1 Report
Reviewing: “Optimum Care of Acute Kidney Injury (AKI) Survivors Post Hospital Discharge”
This paper is a narrative review focusing on the management of patients having suffered of acute kidney injury after there hospital discharge. This represents a growing field of interest with poor data and recommendations. This review needs to be rearranged because the order of the management proposed seems a little wobbly, some references are lacking at my point of view and the English needs to be reviewed.
- Major comments:
· Line 18, the last AKI definition using de KDIGO criteria is now recommended over the RIFLE or AKIN classification. These 3 definitions have not the same acuity to distinguish the morbidity and mortality associated with AKI.
· Line 32, give definitions of AKD, CKD and ESKD; easier told, patients having experienced AKI are at higher risk of declining glomerular filtration, up to ESKD.
· Line 37, the sentence sounds weird (blood catheter infection is not an organ). AKI also affects the brain, the lungs, the gut… cite for example this nice review DOI: 10.1038/ki.2011.241
· I did not find the figure 1.
· Line 48, I don’t understand the purpose of defining an AKI survivor… Survival is a large concept, but the most common concerns are intensive care unit survival, hospital survival and 1 to 5 years survival.
· Line 54, for AKI mortality, you should describe intensive care unit patients, septic shock patients and ward patients, because the mortality is not the same. Talking about a mortality ranging from 6 to 50% is not didactic.
· Line 113 is written the contrary of the conclusion of the article quoted. High albuminuria is always associated with CKD progression.
· I did not find the table 2, all the way, if it is the first table quoted in the manuscript, this should be named “table 1”.
· Line 129, we would attempt to have more details on hemodynamic volume status assessment and goals. References about the most nephrotoxic drugs should be included.
· Line 154, for patients undergoing dialysis after discharge, please mention that the different dialysis technics should be explained to the patients (peritoneal or haemodialysis). And in the hypothesis of future haemodialysis, vascular access should be anticipated with protection of the veins form the non-dominant arm.
· I do not understand why the chapter about patient’s centred outcomes only relates to the dialysis-free patients. This chapter is also confusing, because mortality contrasts with the new considerations of patient’s centred outcomes at my point of view.
· Line 217, the quotation 35-40 refers to articles that have very different patients, setting and time course. Please detail the studies focusing on ACEI and ARBs (brar et al. and DOI: 10.1007/s00134-018-5160-6).
· Line 225, refer at least one time to the recommendation of care after an AKI DOI: 10.1038/nrneph.2017.2
- Minor comments:
· The first sentence of the abstract sounds a little bit childish, at least explain that AKI complicates numbers of diseases and that it is a world wild health problem linked with high morbidity and mortality.
· Line 16, it is defined by a rapid drop in glomerular filtration.
· Line 30, “fluid overload” is more appropriate
· Line 46, no strong data, or trial have proven a survival benefit of adequate care.
· Line 52, there are 2 points at the end of the sentence.
· Line 83, explain what ACEI and ARBs mean.
· Line 102, the reference 22 seems wrong.
· Line 103, you can maybe refer to the article of Ron Wald et al. DOI: 10.1001/jama.2009.1322
· Line 141, education to prevent drug and situation related AKI may work but has limited effects DOI: 10.1016/j.therap.2020.07.004
· Line 149, what is PCP?
· Lines 136 to 153, the part is a little bit redundant.
· Line 176, another strong argument in this way is that in the AKIKI study, patients randomized in the late strategy of KRT had recovered earlier an adequate urine output DOI: 10.1056/NEJMoa1603017
Author Response
Thank you so much for your thorough and thoughtful review. Kindly see my responses below.
Major comments:
- Line 18, the last AKI definition using de KDIGO criteria is now recommended over the RIFLE or AKIN classification. These 3 definitions have not the same acuity to distinguish the morbidity and mortality associated with AKI.
**Changed to use the recommended KDIGI criteria to define AKI
- Line 32, give definitions of AKD, CKD and ESKD; easier told, patients having experienced AKI are at higher risk of declining glomerular filtration, up to ESKD.
**Changed.
- Line 37, the sentence sounds weird (blood catheter infection is not an organ). AKI also affects the brain, the lungs, the gut… cite for example this nice review DOI: 10.1038/ki.2011.241
** Changed and the reference suggested was cited
- I did not find the figure 1.
**Uploaded
- Line 48, I don’t understand the purpose of defining an AKI survivor… Survival is a large concept, but the most common concerns are intensive care unit survival, hospital survival and 1 to 5 years survival.
**As the manuscript is focused on AKI patients post hospital discharge, no details about in-patient factors (though they may affect overall prognosis) were included. We wanted to highlight that we are discussing survival in the outpatient setting and their care..
- Line 54, for AKI mortality, you should describe intensive care unit patients, septic shock patients and ward patients, because the mortality is not the same. Talking about a mortality ranging from 6 to 50% is not didactic.
**As above. Though we agree with reviewer that all these factors may affect the post-hospitalization mortality.
- Line 113 is written the contrary of the conclusion of the article quoted. High albuminuria is always associated with CKD progression.
** Corrected
- I did not find the table 2, all the way, if it is the first table quoted in the manuscript, this should be named “table 1”.
**Uploaded
- Line 129, we would attempt to have more details on hemodynamic volume status assessment and goals. References about the most nephrotoxic drugs should be included.
** The aim of the manuscript is to review the optimum care of AKI patients post hospital discharge, thus we discussed the effect of volume status and ultrafiltration on patients with AKI-D in the outpatient setting quoting Abdel-Rahman et al JCM 2022 and Pajewski et al Hemodial Int 2018.
- Line 154, for patients undergoing dialysis after discharge, please mention that the different dialysis technics should be explained to the patients (peritoneal or haemodialysis). And in the hypothesis of future haemodialysis, vascular access should be anticipated with protection of the veins form the non-dominant arm.
** Agree with reviewer and were added.
- I do not understand why the chapter about patient’s centred outcomes only relates to the dialysis-free patients. This chapter is also confusing, because mortality contrasts with the new considerations of patient’s centred outcomes at my point of view.
** Added a part about patient centred care for patients with AKI_D
- Line 217, the quotation 35-40 refers to articles that have very different patients, setting and time course. Please detail the studies focusing on ACEI and ARBs (brar et al. and DOI: 10.1007/s00134-018-5160-6).
**Done
- Line 225, refer at least one time to the recommendation of care after an AKI DOI: 10.1038/nrneph.2017.2
**Done ( in the introduction)
- Minor comments:
- The first sentence of the abstract sounds a little bit childish, at least explain that AKI complicates numbers of diseases and that it is a world wild health problem linked with high morbidity and mortality.
**Changed
- Line 16, it is defined by a rapid drop in glomerular filtration.
**Changed
- Line 30, “fluid overload” is more appropriate
**Changed
- Line 46, no strong data, or trial have proven a survival benefit of adequate care.
**Agree, we presented our own data.
- Line 52, there are 2 points at the end of the sentence.
**Corrected
- Line 83, explain what ACEI and ARBs mean.
**Done
- Line 102, the reference 22 seems wrong.
**Corrected
- Line 103, you can maybe refer to the article of Ron Wald et al. DOI: 10.1001/jama.2009.1322
**Done
- Line 149, what is PCP?
**Explained
- ·Line 176, another strong argument in this way is that in the AKIKI study, patients randomized in the late strategy of KRT had recovered earlier an adequate urine output DOI: 10.1056/NEJMoa1603017
** Good point, but it is more related with in-patient intervention not post-discharge.

Reviewer 2 Report
This review discusses acute kidney injury, including its complications and the care for acute kidney injury survivors. The following are some comments to enhance the manuscript.
1. The tables and figures cannot be seen in the manuscript which I downloaded, make sure you have uploaded them.
2. Please check if the quotes and references are properly cited in your manuscript, for example:
Line 113-114: please, revisit the following sentence “This study showed that a post-AKI low urine albumin level was associated with increased risk of kidney disease progression, (hazard ratio [HR], 1.53 for each doubling; 95% CI, 1.45-1.62)”. And the sentence in the reference 24 is “Higher post-AKI urine ACR level was associated with increased risk of kidney disease progression (hazard ratio [HR], 1.53 for each doubling; 95% CI, 1.45-1.62).”
Line 83-84: “In patients with perioperative administration of ACEI/ARB and CCB, postoperative AKI was significantly associated with higher risk of fatal stroke and MI, respectively”, which is copied from your reference 19. And the sentence in the reference 19 is “In patients with perioperative administration of ACEI/ARB and CCB, postoperative AKI was significantly associated with higher risk of fatal stroke and MI, respectively.”
3. Please check the abbreviations throughout the text, some appear for the first in the text without being properly spelled out, for example:
Line 35: “HR” and “CI” should be spelled out.
Lines 149: “PCP” is not spelled out.
Line 163: “HD” and “BMP” should be spelled out.
Author Response
Thank you so much for your thorough and thoughtful review. Kindly see my responses below.
This review discusses acute kidney injury, including its complications and the care for acute kidney injury survivors. The following are some comments to enhance the manuscript.
- The tables and figures cannot be seen in the manuscript which I downloaded, make sure you have uploaded them.
** Uploaded.
- Please check if the quotes and references are properly cited in your manuscript, for example:
Line 113-114: please, revisit the following sentence “This study showed that a post-AKI low urine albumin level was associated with increased risk of kidney disease progression, (hazard ratio [HR], 1.53 for each doubling; 95% CI, 1.45-1.62)”. And the sentence in the reference 24 is “Higher post-AKI urine ACR level was associated with increased risk of kidney disease progression (hazard ratio [HR], 1.53 for each doubling; 95% CI, 1.45-1.62).”
Line 83-84: “In patients with perioperative administration of ACEI/ARB and CCB, postoperative AKI was significantly associated with higher risk of fatal stroke and MI, respectively”, which is copied from your reference 19. And the sentence in the reference 19 is “In patients with perioperative administration of ACEI/ARB and CCB, postoperative AKI was significantly associated with higher risk of fatal stroke and MI, respectively.”
**Reviewed and corrected
- Please check the abbreviations throughout the text, some appear for the first in the text without being properly spelled out, for example:
Line 35: “HR” and “CI” should be spelled out.
Lines 149: “PCP” is not spelled out.
Line 163: “HD” and “BMP” should be spelled out.
**Done

Reviewer 3 Report
Dear Authors, after careful review of your article, i did not find major remarks.
Only one minor remark:
on page 3, line 102-104 you repeat an information which was already given on page 1, line 34-35.
Best regards
Author Response
Thank you so much for your thorough and thoughtful review. Kindly see my responses below.
-on page 3, line 102-104 you repeat an information which was already given on page 1, line 34-35.
Best regards
**Corrected.
Round 2
Reviewer 1 Report
Explain ESKD and F/U under the Figure 1
Table 1 is not cited in the text, is it really necessary?
Author Response
My sincere appreciation for you review.
Explain ESKD and F/U under the Figure 1
**Done
Table 1 is not cited in the text, is it really necessary?
** Done
Reviewer 2 Report
Thanks for your correction.
Author Response
Appreciate your review
You are very welcome